# Modification of *i*-GONAD Suitable for Production of Genome-Edited C57BL/6 Inbred Mouse Strain

**DOI:** 10.3390/cells9040957

**Published:** 2020-04-13

**Authors:** Yukari Kobayashi, Takuya Aoshima, Ryota Ito, Ryota Shinmura, Masato Ohtsuka, Eri Akasaka, Masahiro Sato, Shuji Takabayashi

**Affiliations:** 1Laboratory Animal Facilities & Services, Preeminent Medical Photonics Education & Research Center, Hamamatsu University School of Medicine, 1-20-1 Handayama, Higashi-ku, Hamamatsu, Shizuoka 431-3192, Japan; yukari.k.hama@outlook.jp (Y.K.); aoshimat@hama-med.ac.jp (T.A.); A16010@hama-med.ac.jp (R.I.); A17046@hama-med.ac.jp (R.S.); 2Department of Molecular Life Science, Division of Basic Medical Science and Molecular Medicine, Tokai University School of Medicine, Isehara, Kanagawa 259-1193, Japan; masato@is.icc.u-tokai.ac.jp; 3Center for Matrix Biology and Medicine, Graduate School of Medicine, Tokai University, Isehara, Kanagawa 259-1193, Japan; 4The Institute of Medical Sciences, Tokai University, Isehara, Kanagawa 259-1193, Japan; 5Section of Gene Expression Regulation, Frontier Science Research Center, Kagoshima University, Kagoshima 890-8544, Japan; stylistics777@yahoo.co.jp

**Keywords:** low-dose PMSG, genome editing, *i*-GONAD, C57BL/6J, in vivo electroporation, CRISPR/Cas9

## Abstract

Improved genome editing via oviductal nucleic acid delivery (*i*-GONAD) is a novel method for producing genome-edited mice in the absence of ex vivo handling of zygotes. *i*-GONAD involves the intraoviductal injection of clustered regularly interspaced short palindromic repeats (CRISPR) ribonucleoproteins via the oviductal wall of pregnant females at 0.7 days post-coitum, followed by in vivo electroporation (EP). Unlike outbred Institute of Cancer Research (ICR) and hybrid mouse strains, genome editing of the most widely used C57BL/6J (B6) strain with *i*-GONAD has been considered difficult but, recently, setting a constant current of 100 mA upon EP enabled successful *i*-GONAD in this strain. Unfortunately, the most widely used electroporators employ a constant voltage, and thus we explored conditions allowing the generation of a 100 mA current using two electroporators: NEPA21 (Nepa Gene Co., Ltd.) and GEB15 (BEX Co., Ltd.). When the current and resistance were set to 40 V and 350–400 Ω, respectively, the current was fixed to 100 mA. Another problem in using B6 mice for *i*-GONAD is the difficulty in obtaining pregnant B6 females consistently because estrous females often fail to be found. A single intraperitoneal injection of low-dose pregnant mare’s serum gonadotrophin (PMSG) led to synchronization of the estrous cycle of these mice. Consequently, approximately 51% of B6 females had plugs upon mating with males 2 days after PMSG administration, which contrasts with the case (≈26%) when B6 females were subjected to natural mating. *i*-GONAD performed on PMSG-treated pregnant B6 females under conditions of average resistance of 367 Ω and average voltage of 116 mA resulted in the production of pregnant females at a rate of 56% (5/9 mice), from which 23 fetuses were successfully delivered. Nine (39%) of these fetuses exhibited successful genome editing at the target locus.

## 1. Introduction

Recently, clustered regularly interspaced short palindromic repeats (CRISPR)/caspase-9 (Cas9) (CRISPR/Cas9) gene editing technology has been widely employed for the rapid generation of genetically modified (GM) animals, due to its simplicity, versatility, and efficiency [1,2]. This technology has enabled the production of mice with knockout, knock-in (KI), or conditional alleles, or those with single point mutations, within a few weeks. For the generation of GM animals, many researchers employed the microinjection of CRISPR components into zygotes using a micromanipulator system or in vitro electroporation (EP) of zygotes in a solution containing the CRISPR components using an electroporator [3,4,5,6,7,8,9,10,11,12,13,14,15]. These processes require ex vivo handling of zygotes and embryo transfer of the genome editing-treated zygotes to the reproductive tract of a recipient female for further development, all of which are costly, time-consuming, and labor-intensive.

We developed a new method, called genome editing via oviductal nucleic acid delivery (GONAD), which was subsequently renamed “improved GONAD (*i*-GONAD)”, for the production of genome-edited mice [16,17,18], rats [19,20], and hamsters [21]. This technology is based on the injection of a solution (1–1.5 μL) containing genome editing reagents into the lumen of an oviduct via the oviductal wall of pregnant female animals at the late zygote to two-cell stage following in vivo EP of the entire oviduct using tweezer-type electrodes under a dissecting microscope [22,23]. In other words, in the case of GONAD/*i*-GONAD, genome editing occurs in preimplantation embryos floating in the oviductal lumen in situ. The genome-edited embryos will develop into fetuses after implantation, and the fetuses will be delivered naturally as genome-edited newborn animals. Thus, this technology does not require ex vivo handling of embryos, which is strictly required for the above-mentioned zygote microinjection and in vitro EP. In this context, GONAD/*i*-GONAD appears to be more convenient and simpler than the methods that are based on the ex vivo handling of embryos.

According to Ohtsuka et al. [17] and Gurumurthy et al. [22], GONAD/*i*-GONAD is easier to use on hybrid mice such as B6C3F1 (a hybrid between C57BL/6 and C3H/He) and randomly bred Institute of Cancer Research (ICR) mice than on the widely used inbred strain C57BL/6 (B6). Specifically, the electric conditions used for hybrid mice and ICR were often found to be deleterious to the development of B6 embryos. Ohtsuka et al. [17] demonstrated successful acquisition of genome-edited B6 offspring when *i*-GONAD was carried out using the CUY21EditII electroporator provided by BEX Co., Ltd. (Tokyo, Japan), which has the capacity to generate a constant current. According to Gurumurthy et al. [22], the optimal electric conditions for B6 involve in vivo EP being carried out under a constant current of 100 mA. To our knowledge, two electroporators, NEPA21 (NEPA GENE Co. Ltd., Chiba, Japan) and CUY21EditII, are those that have been most frequently used for successful GONAD/*i*-GONAD [16,17,18,19,20,21]. CUY21EditII can provide a constant current, whereas NEPA21 cannot. In this study, we attempted to explore the electric conditions (resistance and voltage) that provide a current of 100 mA when the NEPA21 electroporator is used.

Another problem in using B6 mice for *i*-GONAD is the difficulty in consistently obtaining pregnant B6 females because estrous females often fail to be found. The administration of gonadotrophins has been used for inducing superovulation in many mouse strains to obtain a number of early embryos [24]. This approach has an additional advantage in that it is capable of synchronizing the estrous cycle of females; thus, it is unnecessary to check the estrous cycle through smear testing or to inspect the state of the vagina visually. Sato et al. [25] reported that intraperitoneal (IP) administration of low-dose (0.02–0.2 international units (IU)) pregnant mare’s serum gonadotrophin (PMSG) following IP administration of 5 IU human chorionic gonadotrophin (hCG) 48 h apart is effective for natural ovulation induction before *i*-GONAD. Unfortunately, this regime has only been used successfully on B6C3F1 hybrid mice. In this study, we examined whether the administration of a single IP injection of low-dose PMSG is effective for synchronizing the estrous cycle in B6 females.

## 2. Materials and Methods

### 2.1. Animals

C57BL/6JJmsSlc (B6), BALB/cCrSIc (BALB/c), and Slc:ICR (ICR) mice were purchased from Japan SLC, Inc. (Shizuoka, Japan). Adult mice at the age ranges of 8–10 weeks (female) and 10–12 weeks (male) were used. All mice were maintained under temperature-controlled conditions (24 ± 2 °C) with a 12L/12D light cycle (lights on at 7:00 a.m.). A solid diet and water were provided ad libitum. The experiments described were performed in accordance with the guidelines of Hamamatsu University School of Medicine Committee on Recombinant DNA Security (No. 29–17). Additionally, the experiments were approved by Hamamatsu University School of Medicine Animal Care and Use Committee (No. 2017097). The experiments involving in vivo transfection of mouse preimplantation embryos by *i*-GONAD were accompanied by surgery (exposure of ovary/oviducts/uterus) and operation/manipulation (DNA injection via the oviductal wall and in vivo EP). All efforts were made to minimize the number of animals used and their suffering.

### 2.2. Mating Protocol

In the natural mating group, each female mouse was coupled with two males in a cage in the evening. In the PMSG treatment group, female mice received an IP injection of 1.2 IU PMSG/10 g in the evening (5:00–6:00 p.m.). Then, 47–49 h later, each treated female mouse was mated with two males in a cage. The morning after mating, females were checked for the presence of a copulation plug to confirm that mating had occurred. The plugged females were subjected to *i*-GONAD or remained untreated until birth to evaluate the following pregnancy indices: plug rate (number of plugged females/number of mated females × 100%), pregnancy rate (number of pregnant females/number of plugged females × 100%), and size of first litters (number of pups/number of females delivering live pups).

### 2.3. Measurement of Current

We used two types of square-pulse generator: NEPA21 (NEPA GENE Co. Ltd. Chiba, Japan) and GEB15 (BEX Co. Ltd., Tokyo, Japan). NEPA21 generates a poring pulse (Pp) and a transfer pulse (Tp). The electroporation parameters were as follows: 3 Pp (40 V, 5 ms wavelength, 50 ms pulse interval, 10% decay (±pulse orientation)) and 3 Tp (10 V, 50 ms wavelength, 50 ms pulse interval, 40% decay (±pulse orientation)). For GEB15, the electroporation parameters were as follows: driving pulse voltage (Pd V): 40 V, Pd on: 5.00 ms, Pd off: 50 ms, pulse cycles: 3, and decay: 10%. Notably, the voltage was set to 40 V (Pp for NEPA21 and PdV for GEB15).

Non-pregnant adult B6 females were euthanized with a mixture of three anesthetic agents (medetomidine, midazolam, and butorphanol) and incised at the dorsal skin for exposure of the ovary/oviduct/uterus. The oviduct regions were then covered with a piece of wet paper (KimWipe; Jujo-Kimberly Co. Ltd., Tokyo, Japan) soaked in Dulbecco’s modified phosphate-buffered saline (DPBS) and then sandwiched in tweezer-type electrodes, CUY652-3 (NEPA GENE Co. Ltd.). In this case, the interval between the two electrodes varied from 0 to 2 mm (Appendix A). First, the resistance value was measured by pressing the Ω button set in each electroporator when the oviduct was sandwiched with the electrodes. Next, immediately after measuring the resistance, the start button was pressed to check the current.

### 2.4. Preparation of Genome Editing Reagents Used for i-GONAD

For the CRISPR/Cas9-mediated induction of insertions and deletions (indels) in B6, Alt-R CRISPR-Cas9 CRISPR RNAs (crRNAs) were designed to recognize the target site (exon 1 of wild mouse tyrosinase gene (*Tyr*) that matches a 20 bp DNA sequence (Tyr-wild crRNA: GGAAACTGTAAGTTTGGATT) just upstream of the protospacer adjacent motif (PAM). crRNAs were synthesized by Integrated DNA Technologies, Inc. (IDT) (Coralville, IA, USA). Alt-R CRISPR-Cas9 trans-activating small RNA (tracrRNA) was also purchased from IDT. Each reagent was dissolved in Opti-MEM medium (#31985062; Thermo Fisher Scientific Inc., Waltham, MA, USA) at a final concentration of 100 μM, and stored at −80 °C until use. crRNAs and tracrRNA were annealed by mixing equimolar amounts of each and incubating at room temperature for about 10 min to allow the formation of crRNA/tracrRNA duplexes. The annealed crRNA/tracrRNA duplexes (6 μL) were mixed with 1 μL of Cas9 protein (10 μg/μL; purchased from IDT) to form ribonucleoprotein (RNP) complexes in 2 μL of Opti-MEM medium in a 0.5-mL PCR tube. Thus, the solution used for *i*-GONAD comprised crRNA/tracrRNA duplexes (final concentration of 30 μM), Cas9 protein (1 μg/μL), and 0.02% Fast Green FCF (#15939-54; Nacalai Tesque Inc., Kyoto, Japan) dissolved in Opti-MEM medium.

For CRISPR/Cas9-mediated KI using a single-stranded oligodeoxynucleotide (ssODN), the Tyr-ssODN (5′-TCT GTG TTT TAT AAT AGG ACC TGC CAG TGC TCA GGC AAC TTC ATG GGT TTC AAC TGC GGA AAC TGT AAG TTT GGA TTT GGG GGC CCA AAT TGT ACA GAG AAG CGA GTC TTG ATT AGA AGA AAC ATT TTT G-3′) with 64 bp (left arm) and 65 bp (right arm) homologous arms at both ends was custom-synthesized (Macrogen, Seoul, South Korea). The lyophilized ssODN was re-suspended in Opti-MEM to a concentration of 10 µg/µL. Alt-R CRISPR-Cas9 crRNAs were designed to recognize the target site (exon 1 of albino mouse *Tyr* (Figure 3a and Figure 4a)) that matched a 20 bp DNA sequence (Tyr-albino crRNA: GGAAACTCTAAGTTTGGATT) just upstream of the PAM [17]. Annealed crRNA/tracrRNA duplexes (3 μL each) were mixed with Cas9 protein (1 μL) and ssODN (2 μL) in Opti-MEM to a total of 10 μL in a 0.5 mL PCR tube, so that the final concentrations of components were 30 µM crRNA/tracrRNA duplexes, 1 μg/μL Cas9 protein, 2 µg/µL ssODN, and 0.02% Fast Green FCF.

### 2.5. i-GONAD Procedure

We performed *i*-GONAD at 0.7 days post-coitum (dpc) (4:00 p.m.). Surgical procedures were performed on adult females anesthetized with a mixture of three anesthetic agents (medetomidine (0.75 mg/kg; Nippon Zenyaku Kogyo Co. Ltd., Fukushima, Japan), midazolam (4 mg/kg; Sandoz K.K., Tokyo, Japan), and butorphanol (5 mg/kg; Meiji Seika Pharma Co., Ltd., Tokyo, Japan)) under a dissecting microscope, on the basis of a slightly modified versions of previously reported procedures [17,22,23]. The ovary/oviduct/uterus were exposed after making an incision (≈1 cm in length) in the dorsal skin and a subsequent incision (≈0.5 cm in length) in the muscle layer. The exposed ovary, oviduct, as well as part of the uterus, were placed on the back skin of mice, and adipose tissue was anchored with an Aorta-Klemme to prevent return of the exposed tissues. Approximately 1.0 μL of solution was injected into the oviduct lumen upstream of the ampulla using a micropipette (prepared using an electric puller (#PN-3; NARISHIGE Co. Ltd., Tokyo, Japan)) and an attached mouthpiece. Immediately after the injection, the oviduct regions were covered with a piece of wet KimWipe paper, soaked in DPBS, and grasped in tweezer-type electrodes (#CUY652-3; NEPA GENE Co. Ltd.). Electroporation was performed using a square-wave pulse generator: NEPA21 (NEPA GENE Co. Ltd.) or GEB15 (BEX Co. Ltd.). The electroporation parameters were as follows, based on previous papers [22,23]: 3 Pp (50 V/5 ms wavelength/50 ms duration/10% decay rate/± polarity) and 3 Tp (10 V/50 ms wavelength/50 ms duration/40% decay rate/± polarity) for NEPA21 and Pd; V: 40 V or 60 V, Pd on: 5.00 ms, Pd off: 50 ms, Pd N: 3/10% decay for GEB15. After EP, oviducts were returned to their original position, the incisions made in the internal dorsal muscle were sutured, and the dorsal skin was closed using a surgical stapler. The animals were then given an intradermal detoxicant, atipamezole (3.75 mg/kg; ANTISEDAN, Nippon Zenyaku Kogyo Co. Ltd., Fukushima, Japan), monitored for anesthesia recovery, and housed for further analysis.

### 2.6. Analysis of CRISPR/Cas9-Induced Indels

Pregnant female mice at 12.5 or 13.5 dpc were subjected to euthanasia. The fetuses were dissected out in DPBS. Pigmentation of the eyes and morphological abnormalities were assessed under a dissecting microscope and photographed. Tail biopsies were then taken for the isolation of genomic DNA from the fetuses. Genomic DNA was isolated from the tail biopsies by incubation in 100 μL of 50 mM NaOH at 95 °C for 10 min. Then, 10 μL of 1 M Tris-HCl (pH 8.0) was added to the aliquot and mixed. This crude DNA extract was used as a template for PCR.

PCR amplification of a sequence corresponding to *Tyr* was performed in a volume of 20 µL containing 10 µL of 2× PCR buffer for KOD FX (TOYOBO, Osaka, Japan), 0.4 mM deoxyribonucleotides (dNTPs) mix, 1 µL of crude lysate, 0.25 µM primer pairs (Tyr–F: TCT CTG ATG GCC ATT TTC CTC/Tyr–R: AAC ATG GGT GTT GAC CCA TT) [17], and 0.1 U KOD FX (TOYOBO, Osaka, Japan) under cycling conditions of denaturation at 94 °C for 3 min; amplification with 33 cycles of 95 °C for 20 s, 57 °C for 30 s, and 68 °C for 1 min; and extension at 68 °C for 5 min. Amplification products (5 μL) were separated by 2% agarose gel electrophoresis.

Direct sequencing was performed using the PCR products and the primer Tyr–F with a BigDye Terminator v3.1 Cycle Sequencing kit (Thermo Fisher Scientific Inc.), and then analyzed on an automated ABI PRISM 3100 DNA sequencer (Thermo Fisher Scientific Inc.).

## 3. Results

### 3.1. Determination of Optimal Current

According to Gurumurthy et al. [22], *i*-GONAD under a constant current of 100 mA led to the successful production of genome-edited B6 mice. Unfortunately, presently available electroporators, as exemplified by NEPA21, employ a constant voltage. It is theoretically possible to determine the electric conditions corresponding to 100 mA by measuring the resistance when a constant voltage is set. To determine the optimal electric conditions of NEPA21 suitable for *i*-GONAD-based genome editing in B6 mice, we used two electroporators, NEPA21 and GEB15. Prior to measurement of the resistance, the voltage of all of these electroporators was set to 40 V (for Tp in the case of NEPA21; for PdV in the case of GEB15).

Ovary/oviduct/uterus were pulled out and placed on the back skin of a non-pregnant B6 female, and then the oviduct was covered with a small piece of wet KimWipe paper. In this case, no injection towards the oviduct was performed. The covered oviduct was then sandwiched by tweezer-type electrodes at intervals varying from 0 to 2 mm (see Appendix A). Next, the resistance was measured through pressing the Ω button on each electroporator. On the basis of Ohm’s law, the current can be determined using the following formula: I(A) = 40(V)/R(Ω). In Figure 1, the relationship between the resistance and current obtained through this approach is shown. Each dot in Figure 1 corresponds to a single trial for measurement. In each electroporator, when the resistance ranged from 350 to 400 Ω, the current was evaluated to be ≈100 mA.

### 3.2. i-GONAD-Based Indel Induction Using B6 Mice

When we performed *i*-GONAD using B6 mice, the rate of successfully producing genome-edited mice was extremely low when NEPA21 was used as an electroporator [17]. At this time, we did not check the electric conditions. To explore the cause of this failure, we performed *i*-GONAD (targeted to the murine tyrosinase gene (*Tyr*) as shown in Figure 2a) again using the same conditions (Tp: 40 V; Ω: 100–200 Ω) as previously applied to examine the current and the rate of successful genome editing. Unfortunately, we failed to obtain viable fetuses despite a total of 11 trials. In this experiment, the resistance values ranged from 100 to 215 Ω, yielding an average of 162 Ω, and the current values ranged from 210 to 579 mA, yielding an average of 313 mA (Table 1 and Appendix A). *i*-GONAD under the current of 210 mA resulted in no pregnant females (Appendix A). Inspection was performed 2 days after *i*-GONAD revealed that almost all embryos (morulae) had died or were showing developmental arrest (Appendix A).

Next, we performed *i*-GONAD using NEPA21 with plugged B6 females (obtained after natural mating) under electric conditions in which the resistance values had been adjusted to 350–400 Ω, which may correspond to ≈100 mA. Through a total of 10 trials, four females (40%; termed B6-1, -3, -23, and -66) became pregnant. They had 21 fetuses (with 5.3 fetuses on average), of which a fetus with eyes lacking pigment (shown by T92 in Figure 2b) was found. This phenotype suggested the presence of mutations in both alleles for *Tyr*. Direct sequencing of the PCR products derived from the T92 sample indeed confirmed this (T92 in Figure 2c). The fetus numbered T3 showed no phenotypic alteration with pigmented eyes (T3 in Figure 2b), but sequencing revealed the presence of a mutation in one allele of *Tyr* (T3 in Figure 2c). Of 21 fetuses obtained, 12 (57%) had indels in *Tyr* (Table 1 and Appendix A): three (25%) were judged as having bi-allelic mutations for *Tyr*, four (33%) as having mono-allelic mutations, and five (42%) as having mosaic mutations. The actual resistance values and current values for each pregnant female are shown in Appendix A.

### 3.3. i-GONAD-Based KI Induction Using BALB/c Mice

The production of genome-edited mice of the BALB/c strain was also extremely poor when NEPA21 was used as an electroporator [17]. To examine whether the current can also be an important factor for determining the successful production of genome-edited BALB/c fetuses, we performed *i*-GONAD-based KI. BALB/c mouse has depigmented eyes, as it has a mutation in exon 1 of *Tyr* (Figure 3a). For this experiment, we used gRNA targeted to a sequence spanning a mutation site (C) on exon 1 of *Tyr* and 130 bp oligodeoxynucleotides (ODNs; containing normal nucleotide G at the mutated site) as a donor (Figure 3a).

At first, *i*-GONAD was performed on plugged BALB/c females (obtained after natural mating) under electric conditions of Tp of 40 V and resistance of 100–200 Ω using the NEPA21 electroporator, the same as how the initial *i*-GONAD was performed using B6 mice. Unfortunately, we failed to obtain viable fetuses from the seven females tested. The resistance ranged from 108 to 186 Ω (131 Ω on average), and the current ranged from 205 to 526 mA (368 mA on average) (Table 1 and Appendix A). Next, we performed *i*-GONAD using the electric conditions of Tp of 40 V and resistance of 350–400 Ω with the NEPA21 electroporator. Three (21%; BALB-5, -18, and -23) of 14 BALB/c females had a total of 16 mid-gestational fetuses (5.3 on average) (Table 1). Two (13%) of these 16 fetuses showed pigmented eyes (as exemplified by the fetus numbered T16 in Figure 3b), suggesting successful KI in at least one allele of *Tyr*. Direct sequencing analysis confirmed this (T16 in Figure 3c). The fetus numbered T13 showed no phenotypic alteration with depigmented eyes (T13 in Figure 3b), but sequencing revealed the presence of indels in *Tyr* in both alleles (T3 in Figure 2c). Direct sequencing analysis of the PCR products derived from a total of 16 fetuses demonstrated that 8 (50%) fetuses had KI or indels at the target *Tyr* (Table 1 and Appendix A). The actual resistance values and current values for each pregnant female were shown in Appendix A.

### 3.4. Effect of Electric Conditions on Fetal Development in B6 and BALB/c Mice

As shown previously, we succeeded in obtaining genome-edited fetuses from the B6 and BALB/c females after *i*-GONAD using the modified electric conditions with the NEPA21 apparatus. Unfortunately, in both strains, the rates of fetuses recovered from the *i*-GONAD-treated females appeared to be low (i.e., 40% for B6 and 21% for BALB/c), despite the females used for *i*-GONAD having been checked for the presence of copulation plugs before the procedure. There was a strain-dependent difference in whether or not plugged females developed fetuses. For instance, in the case of CD-1 mice, an outbred stock, about 70%–75% of plugged females indeed became pregnant, whereas the pregnancy rate was usually below 50% in the case of inbred female mice [26]. To examine whether the low rate of pregnancy in the *i*-GONAD-treated B6 and BALB/c females could be attributed to damage elicited by the electric pulse experienced during *i*-GONAD, we evaluated pregnancy indices (litter size and pregnancy rate of plugged females) using those mice. In the absence of *i*-GONAD, the pregnancy rates of plugged females were 33% (for plugged B6 females (*n* = 9)) and 58% (for plugged BALB/c females (*n* = 12)) (Table 2). Average litter sizes were estimated to be 4.6 (for B6 (*n* = 3 litters) and 8.4 (for BALB/c (*n* = 7 litters)) (Table 2). On the basis of data showing that the pregnancy rate of plugged females after *i*-GONAD was 40% for B6 females and 21% for BALB/c females, *i*-GONAD appeared not to affect either the pregnancy rate or the litter size of B6 (but not BALB/c) mice. Regarding BALB/c mice, the present electric conditions used for *i*-GONAD may still not be optimal, although we obtained several genome-edited individuals.

### 3.5. i-GONAD Using GEB15 Electroporator

GEB15 is also a widely used electroporator with the ability to generate a constant voltage. To our knowledge, no reports have been published about the use of this machine for *i*-GONAD. As such, this attempt to perform *i*-GONAD using GEB15 may be the first of its kind. Plugged ICR females were subjected to *i*-GONAD through intraoviductal injection of a solution containing Cas9/gRNA complex targeted to exon 1 of *Tyr* and ssODN donor (all of which have been used for KI in BALB/c mice) and subsequent in vivo EP (60 V/250–400 Ω/130–335 mA). A total of 11 out of 14 females became pregnant and finally delivered a total of 41 fetuses. Of these fetuses, 17 (42%) exhibited pigmentation in their eyes, suggesting successful KI (Tc54 in Figure 4b), which was confirmed by direct sequencing analysis (Tc54 in Figure 4c). The fetus numbered Tc64 showed no phenotypic alteration with depigmented eyes (Tc64 in Figure 4b), but sequencing revealed the presence of indels in *Tyr* in both alleles (Tc64 in Figure 4c). Direct sequencing analysis of the PCR products derived from the fetuses having depigmented eyes demonstrated that 29 (71%) of 41 fetuses obtained could be identified as genome-edited fetuses with either KI or indels at the target *Tyr* (Appendix A). Thus, GEB15 can be used for *i*-GONAD-based genome editing, at least in outbred mice such as ICR.

Next, to examine whether GEB15 can be used for *i*-GONAD-based genome editing in B6 mice, we used electric conditions in which the current of ≈100 mA was adjusted by changing the resistance from 300 to 450 while fixing the voltage at 40 V, as was the case when *i*-GONAD was performed using NEPA21. Intraoviductal injection of a solution containing Cas9/gRNA targeted to exon 1 of *Tyr* was first performed on pregnant B6 females and subsequently the injected oviducts were subjected to in vivo EP under the above electric conditions. Of 10 B6 females used, 4 (40%; B6-37, -44, -45, and -50) successfully delivered a total of 21 fetuses with an average litter size of 5.3. Of these 21 fetuses, 10 (48%) were identified as having indels (Table 1 and Appendix A). Of these 10 genome-edited fetuses, 6 (29%) had bi-allelic mutations and 4 (19%) had mono-allelic mutations (Appendix A). The actual resistance values and current values for each pregnant female are shown in Appendix A. Notably, the pregnancy rate of plugged females and average litter size obtained after *i*-GONAD were comparable to those obtained when *i*-GONAD was performed using NEPA21.

### 3.6. Effects of A Single Administration of Low-Dose PMSG on the Total Performance of i-GONAD

Besides exploring the optimal electric conditions for *i*-GONAD in B6 and BALB/c mice, there is another issue concerning the judgment of whether females are in the estrous cycle before they are mated with males. Administration of low-dose gonadotrophins may resolve this issue, because this is known to synchronize the estrous cycle in mice.

First, B6 females were IP injected with 2.4 IU PMSG in the evening (5:00–6:00 p.m.). Forty-eight hours later, the females were mated with males. The next morning, the presence of copulation plugs was inspected. In the case of mating of hormone-uninjected females with males, 16 (26%) of 62 females were successfully found to have plugs (Table 3). In contrast, when hormone-injected females were mated with males, 25 (51%) out of 49 tested had plugs, showing approximate doubling of the rate of successful mating (Table 3). Four (44%) of nine hormone-injected and plugged females delivered pups with an average litter size of six, which was comparable to the rate obtained by hormone-uninjected females. Similar treatment was performed on BALB/c mice. In the case of mating of hormone-uninjected females with males, 21 (35%) of 60 females successfully showed plugs (Table 3). In contrast, when hormone-injected females were mated with males, 24 (53%) out of 45 tested had plugs, showing a rougly 1.5-fold increase in the rate of successful mating (Table 3). A total of 6 (60%) out of 10 hormone-injected and plugged females delivered pups, which was comparable to the rate obtained for hormone-uninjected females (Table 2). These findings indicate that an increased rate of acquiring plugged females is achieved when females are singly administered with low-dose PMSG before mating. This also suggests that it is possible to proceed with scheduled mating to obtain the desired number of plugged females for use for *i*-GONAD.

Second, we examined whether hormone-injected females can deliver genome-edited pups when *i*-GONAD is applied to these mice. Females were first injected with low-dose PMSG, and 48 h later were mated with males. The next day, the plugged females were subjected to *i*-GONAD through intraoviductal injection of a solution containing Cas9/gRNA complex (targeted to exon 1 of *Tyr*) and ssODN donor (all of which have been used for KI in BALB/c mice) and subsequent in vivo EP (40 V/367 Ω (on average)/116 mA (on average)). As a result, five (56%) out of nine B6 females successfully delivered a total of 23 fetuses, and 9 (39%) of these 23 fetuses were identified as having been genome-edited (Table 1 and Appendix A). When a similar approach was performed for BALB/c mice, 4 (36%) out of 11 females delivered 16 fetuses. A total of 8 (50%) of these 16 fetuses were found to have been genome-edited. Four (25%) of these eight genome-edited fetuses had successful KI (Table 1 and Appendix A).

From these findings, in both B6 and BALB/c mice, the number of pregnant females increased when low-dose hormone was added before mating, although litter size remained unchanged between before and after hormone administration. Notably, *i*-GONAD reduced the number of pregnant BALB/c females despite hormone injection: for example, 36% of females became pregnant after *i*-GONAD, whereas 60% of females did in the absence of *i*-GONAD. This suggests that, for *i*-GONAD-targeted hormone-treated BALB/c females, the electric conditions might still be suboptimal, such as in the case of hormone-untreated BALB/c mice.

## 4. Discussion

*i*-GONAD is now recognized as an efficient and convenient method for producing genome-edited mice such as ICR and hybrid mice. It strictly requires an electroporator, a square pulse-generating apparatus. However, it has been difficult to apply the technology to inbred strains, as exemplified by the B6 strain, which has been widely used as experimental mice [17]. To overcome this issue, Gurumurthy et al. [22] focused on the use of a constant current upon *i*-GONAD, and found that *i*-GONAD performed using an electroporator capable of generating a constant current of 100 mA was suitable for acquiring genome-edited B6 mice. Notably, this was in contrast with the case of ICR mice, which usually required 100–200 mA for their genome editing upon *i*-GONAD [22]. Unfortunately, the electroporators most widely used appear to be those capable of generating a constant voltage. To adapt these apparatuses for *i*-GONAD using B6 mice, we here explored electric conditions enabling the generation of a current of 100 mA through changing the resistance (which can be achieved by changing the distance between the two electrodes when an intact oviduct is sandwiched by those electrodes). When we used two electroporators driven under a constant voltage, NEPA21 and GEB15, they both exhibited current values on the basis of Ohm’s law, although in some cases a high current (which deviated from Ohm’s law) was observed (see Figure 1). We thus examined the resistance (R or Ω) in the case with a voltage fixed at 40 V, as the current can be easily determined through the following formula: R (Ω) = −40 V/0.1 A. Our results showed that, under resistance of 400 Ω, a current of ≈100 mA is generated. When *i*-GONAD was performed using NEPA21 under the conditions in which the resistance varied from 350 to 400 Ω, genome-edited B6 and BALB/c mice were successfully obtained (see Table 1, Appendix A; Figure 2 and Figure 3). In contrast, under more stringent conditions (40 V/100–200 Ω/ ≈300 mA), which are known to be efficient for successfully producing genome-edited mice such as ICR and other hybrid mice [17,22], there were no fetuses from the B6 and BALB/c strains (see Table 1, Appendix A). From these experiments, *i*-GONAD employing a 100 mA current was found to be important for these inbred strains.

Notably, it still remains a problem that the pregnancy rate of the *i*-GONAD-treated B6 females appears to be low. There are several reports describing the fact that the pregnancy rate of plug-positive B6 females ranged from 35% to 85%, which clearly varied among the laboratories tested [26,27]. To examine whether our success in the *i*-GONAD-based production of genome-edited fetuses was a result of appropriate in vivo EP or was simply related to the pregnancy rate of B6 mice themselves, we checked the pregnancy rate of intact B6 females as well as their litter size. When B6 mice were subjected to *i*-GONAD, the rate of successful genome editing in fetuses was 40% and the average litter size was 5.3 (see Table 1). On the other hand, the pregnancy rate of intact B6 females was 33% and their average litter size was 4.6 (see Table 2), suggesting no appreciable difference between these two groups. This in turn means that the electric conditions employed here appear to be optimal for the *i*-GONAD-based production of genome-edited B6 mice. In the case of BALB/c mice, the pregnancy rate of intact females was 58% and their average litter size was 8.4 (see Table 2); however, the pregnancy rate of the *i*-GONAD-treated females was 21% and their average litter size was 5.3 (see Table 1). This means that the electric conditions employed here appear to still be suboptimal for generating genome-edited BALB/c mice using *i*-GONAD. In this context, further exploration is needed with regards to the optimal conditions enabling increases of the pregnancy rate and litter size of the *i*-GONAD-treated BALB/c females.

According to previous reports [16,17,22], the available electroporators used for *i*-GONAD were BTX T-820 (Harvard Apparatus, Inc., Holliston, MA, USA), NEPA21, and CUY21EditII. The GEB15 apparatus is a cheap edition of CUY21EditII and employs a constant voltage. Unlike NEPA21, GEB15 generates a driving pulse (corresponding to Tp) alone. Thus, GEB15 has been considered as a machine specifically for use for in vitro EP. We considered the fact that GEB15 is also useful for in vivo EP like NEPA21 if the resistance is appropriately adjusted. When plugged ICR females were subjected to *i*-GONAD using GEB15, we successfully obtained genome-edited fetuses with relatively high efficiency (71%) (see Table 1 and Appendix A; Figure 4). Furthermore, GEB15 was found to be applicable to the production of genome-edited B6 individuals when the current was set to ≈100 mA. More surprisingly, the genome editing efficiency achieved by using GEB15 was comparable to that achieved by using NEPA21 (see Table 1). It is thus conceivable that any type of electroporator currently or previously commercially available can be fitted to *i*-GONAD if the voltage and resistance are appropriately adjusted.

It is generally difficult to perform planned breeding in B6 mice. In contrast with the case of albino mice such as ICR, which are easy to recognize as being suitable for mating through visual inspection of the vagina, it is difficult to use this approach on B6 females. Recently, Sato et al. [25] demonstrated the usefulness of low-dose (0.02–0.2 IU) administration of PMSG, a gonadotrophic hormone widely used for inducing superovulation in experimental mice, to simultaneously prepare a number of plugged females. Unfortunately, this approach was only confined to hybrid mice such as B6C3F1, and employed hCG, a gonadotrophic hormone, to stimulate ovulation. We here examined whether this approach is applicable to B6 females to examine pregnancy indices such as the rate of plugged females, pregnancy rate of plugged females, and first litter size. In this case, we modified the protocol [25] and employed a single administration of low-dose (1.2 IU) PMSG alone. When B6 or BALB/c females were mated with males 2 days after the administration of PMSG, the rates of plugged females in the PMSG-treated group were increased 2- or 1.5-fold (for B6 and BALB/c, respectively) compared with those in the PMSG-untreated group. This means that about 50% of females became successfully plugged after PMSG treatment. The pregnancy rate of plugged females slightly increased in both strains after PMSG treatment (see Table 2). The first litter size increased in B6 females but decreased slightly in BALB/c females after PMSG treatment (see Table 2). These findings support the usefulness of low-dose PMSG as a convenient way of achieving planned breeding. When this approach was combined with *i*-GONAD, the pregnancy rates of B6 and BALB/c females increased 1.6- and 1.7-fold, respectively (see Table 1). Thus, it is concluded that the administration of low-dose PMSG is beneficial for the *i*-GONAD-based production of genome-edited mice.

## 5. Conclusions

In this study, we explored optimal electric conditions that allow the *i*-GONAD-based production of the genome-edited B6 strain, as well as other inbred strains such as BALB/c when a universal electroporator (driven under a constant voltage) such as NEAP21 and GEB15 is employed. We found that zygotes from those inbred strains could be successfully genome-edited without overt developmental damage when *i*-GONAD was performed under the electric conditions of 40 V/350–400 Ω/100 mA. Furthermore, we found that low-dose PMSG was effective for simultaneously acquiring a number of plugged females. Consequently, the pregnancy rates of B6 and BALB/c mice after *i*-GONAD treatment increased 1.6- and 1.7-fold, respectively, compared with those of PMSG-untreated females (see Table 1). This pretreatment before mating is thus beneficial to increase the production rate of genome-edited mice.

Notably, we succeeded in producing genome-edited rats (including Sprague-Dawley (SD), Lewis, Brown Norway (BN), and SDBNF1 strains) using *i*-GONAD [20]. Unfortunately, the BN strain was insensitive to this treatment. Given that the BN strain is recognized as one of the standard strains for the Rat Genome Sequencing Consortium (RGSC), the development of a method to generate gene-engineered BN rats is highly desirable [28]. Our previous *i*-GONAD method relied on a constant voltage. If each electric parameter (Ω and mA) is tested under a fixed voltage of 40 V, it may be possible to establish genome-edited BN rats through *i*-GONAD. Our present approach based on checking each electric parameter in pursuit of the optimal *i*-GONAD can also be applied to other inbred strains, as well as wild-derived mice, for which *i*-GONAD has not been considered applicable.

## Figures and Tables

**Figure 1 cells-09-00957-f001:**
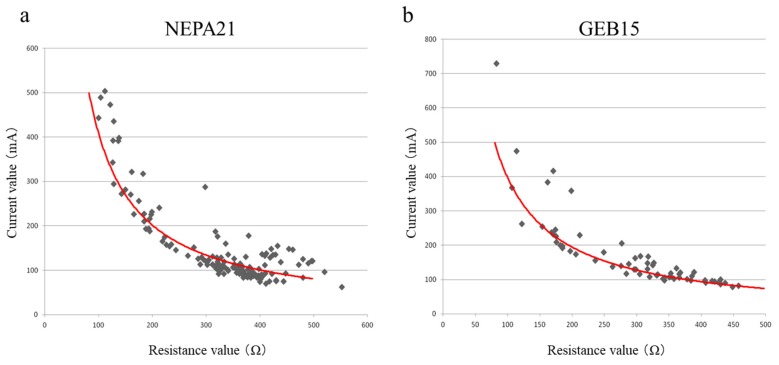
Relationship between the resistance (Ω) and the current (mA) when improved genome editing via oviductal nucleic acid delivery (*i*-GONAD) was performed on intact oviducts of non-plugged B6 females. Two electroporators, NEPA21 (**a**) and GEB15 (**b**), were used for measuring the resistance when the voltage of the pulse (Tp for NEPA21 and PdV for GEB15) was fixed at 40 V. Intact oviduct covered with a small piece of wet KimWipe towel was sandwiched by the tweezer-type electrodes at distances arbitrarily varying from 0 to 2 mm. On the basis of the resulting resistance values, the current was determined. Each dot corresponds to a single trial for measurement. Red lines indicate a curved line obeying Ohm’s law: I(A) = 40(V)/R(Ω).

**Figure 2 cells-09-00957-f002:**
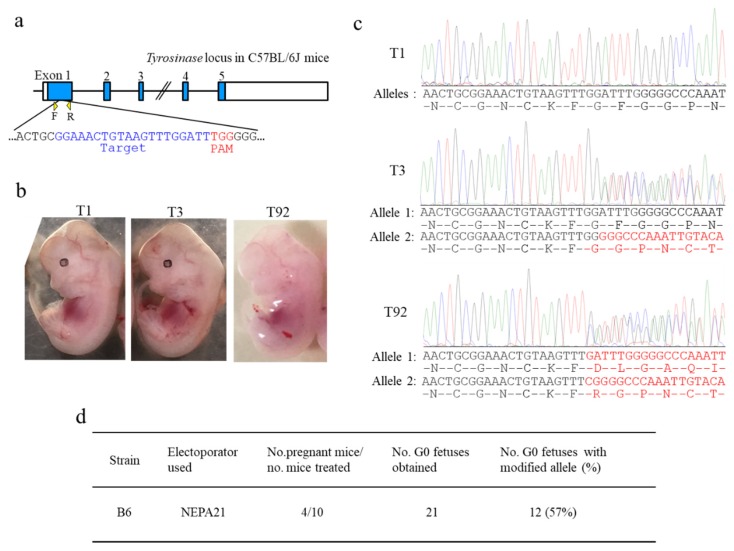
*i*-GONAD-based targeted disruption of tyrosinase gene (*Tyr*) in B6 mice using NEPA21 electroporator under the electric conditions of Tp 40 V, 162 Ω (on average), and ≈100 mA. (**a**) Gene structure of *Tyr* in B6 mice. The target sequence present on exon 1 of *Tyr* and recognized by guide RNA (gRNA) is shown in blue. Primers F and R used for amplification of a region spanning the target sequence are shown below the *Tyr*. Protospacer adjacent motif (PAM) is shown in red. Blue color in each exon corresponds to tyrosinase protein. (**b**) Mid-gestational fetuses (T1, T2, and T92) obtained from *i*-GONAD-treated plug-positive B6 females. Clearly, the T92 fetus exhibited depigmented eyes. (**c**) Direct sequencing of PCR products derived from the fetuses shown in (b). The T1 fetus showing pigmented eyes had wild-type alleles. The T3 fetus showing pigmented eyes is judged as heterozygous mono-allelic knock out (KO) because one allele (called allele 2) had indels, yielding a frameshift of amino acids (shown by red). The T92 fetus is judged as heterozygous bi-allelic KO because both alleles had indels, each yielding different frameshifts of amino acids (shown by red). (**d**) Genome editing efficiency of the *Tyr* locus at B6 strain.

**Figure 3 cells-09-00957-f003:**
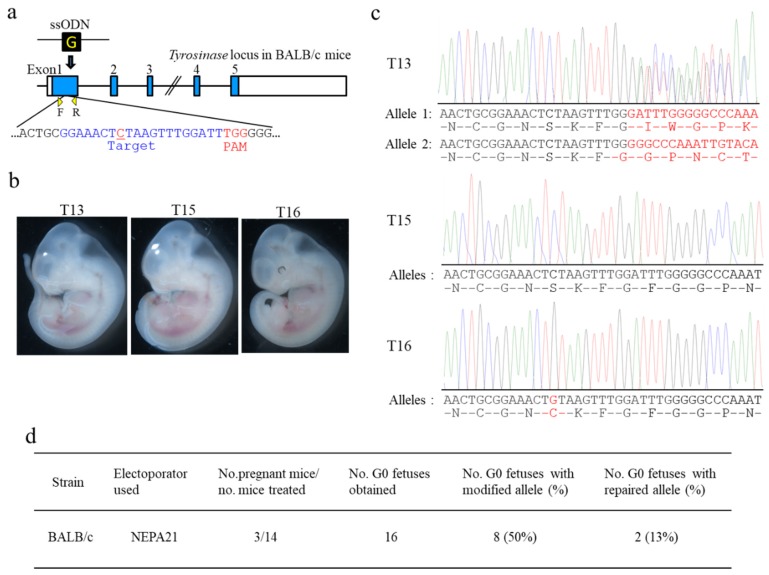
*i*-GONAD-based KI of a wild-type sequence into the tyrosinase gene (*Tyr*) in BALB/c mice using NEPA21 electroporator under the electric conditions of Tp 40 V, 162 Ω (on average), and ≈100 mA. (**a**) Gene structure of *Tyr* in BALB/c mice. BALB/c mouse has a mutated nucleotide “C” (shown in red) in the sequence of exon 1 of *Tyr*. The donor single-stranded oligodeoxynucleotide (ssODN) contains wild-type nucleotide “G” at the same position corresponding to mutated nucleotide “C” because the target nucleotide “G” is a key nucleotide for tyrosinase activity, and thus nucleotide replacement at this position often causes an albino phenotype. Primers F and R used for amplification of a region spanning the target sequence are shown below the *Tyr*. PAM is shown in red. Blue color in each exon encodes tyrosinase protein. (**b**) Mid-gestational fetuses (T13, T15, and T16) obtained from *i*-GONAD-treated plug-positive BALB/c females. Notably, the T16 fetus exhibited depigmented eyes, suggesting successful KI of ssODN. (**c**) Direct sequencing of PCR products derived from the fetuses shown in (b). T15 fetus showing depigmented eyes had mutated alleles. T13 fetus showing depigmented eyes is judged as heterozygous bi-allelic KO because both alleles had indels, each yielding different frameshifts of amino acids (shown by red). The T16 fetus is judged as homozygous bi-allelic KI because both alleles had wild-type nucleotide “G” (shown by red), a key nucleotide for tyrosinase activity. (**d**) Genome editing efficiency of the *Tyr* locus at BALB/c strain.

**Figure 4 cells-09-00957-f004:**
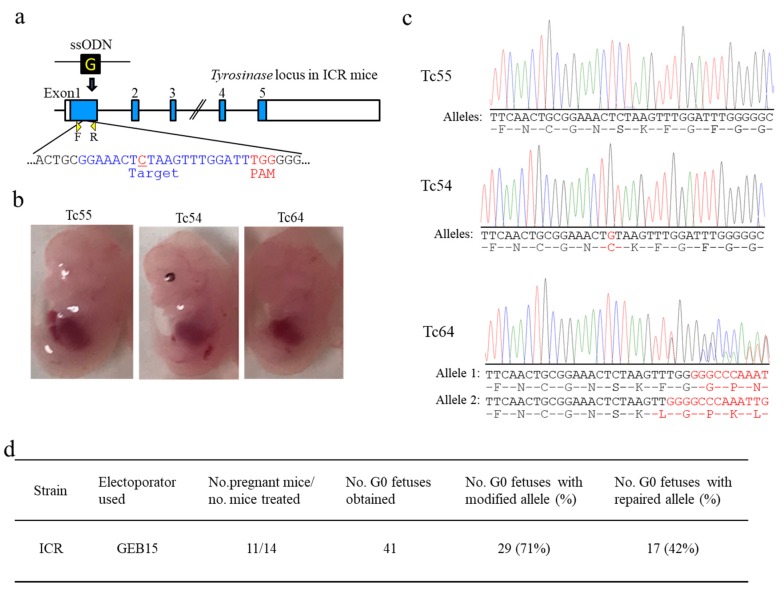
*i*-GONAD-based KI of a wild-type sequence into the tyrosinase gene (*Tyr*) in Institute of Cancer Research (ICR) ice using GEB15 electroporator under the electric conditions of PdV 60 V, 250–400 Ω, and 130–335 mA. (**a**) Gene structure of *Tyr* in ICR mice, which is basically the same as shown in Figure 3a. (**b**) Mid-gestational fetuses (Tc55, Tc54, and Tc64) obtained from *i*-GONAD-treated plug-positive ICR females. (**c**) Direct sequencing of PCR products derived from the fetuses shown in (b). The Tc55 fetus showing depigmented eyes had mutated alleles. The Tc54 fetus showing pigmented eyes is judged as homozygous bi-allelic KI because both alleles had wild-type nucleotide “G” (shown by red), a key nucleotide for tyrosinase activity. The Tc64 fetus showing depigmented eyes is judged as heterozygous bi-allelic KO, each yielding different frameshifts of amino acids (shown by red). (**d**) Genome editing efficiency of the *Tyr* locus at ICR strain.

**Table 1 cells-09-00957-t001:** Summary of *i*-GONAD using two electroporators, NEPA21 and GEB15.

Strain	PMSG Treatment ^1^	Electroporator Used	No. of Plugged Females Subjected to *i*-GONAD	Range of Resistance Value (Ω)	Range of Current Value (mA)	No. of Pregnant Mice (%)	No. G0 Fetuses Obtained (Average Litter Size) ^2^	No. of Modified Fetuses (%) ^3^	No. of Repaired Fetuses (%) ^4^
B6	No	NEPA21	11	100–215	210–579	0	-	-	-
B6	No	NEPA21	10	312–448	74–136	4 (40)	21 (5.3)	12 (57)	-
BALB/c	No	NEPA21	7	108–186	205–526	0	-	-	-
BALB/c	No	NEPA21	14	317–552	62–187	3 (21)	16 (5.3)	8 (50)	2 (13)
B6	No	GEB15	10	299–458	79–149	4 (40)	21 (5.3)	10 (48)	-
ICR	No	GEB15	14	241–476	135–335	11 (79)	41 (3.7)	29 (71)	17 (42)
B6	Yes	NEPA21	9	277–472	75–178	5 (56)	23 (4.6)	9 (39)	-
BALB/c	Yes	NEPA21	11	298–461	75–196	4 (36)	16 (4.0)	8 (50)	4 (25)

^1^ Females were treated with low-dose pregnant mare’s serum gonadotrophin (PMSG) before mating with males, which is defined as “Yes”. Females were not subjected to PMSG treatment before mating with males, which is defined as “No”. In the latter case, females judged as being at the estrous stage, on the basis of visual inspection of the vagina, were selected for mating. ^2^ Average litter size was determined by checking the number of fetuses isolated among the pregnant fetuses. ^3^ No. of modified fetuses indicates those with insertions and deletions (indels) at a target site. ^4^ No. of repaired fetuses indicates those with knock-in (KI) allele(s) at the target site.

**Table 2 cells-09-00957-t002:** Summary of litter size and the rates of females showing successful pregnancy after low-dose PMSG treatment.

Strain	PMSG Treatment ^1^	No. of Plugged Females ^2^	No. of Plugged Females Showing Pregnancy ^3^ (%)	Average First Litter Size ^3^
B6	No	9	3 (33)	4.6
B6	Yes	9	4 (44)	6
BALB/c	No	12	7 (58)	8.4
BALB/c	Yes	10	6 (60)	7.5

^1^ Females were treated with low-dose PMSG before mating with males, which is defined as “Yes”. Females were not subjected to PMSG treatment before mating with males, which is defined as “No”. In the latter case, females judged as being at the estrous stage, on the basis of visual inspection of the vagina, were selected for mating. ^2^ Presence of plugs at the vagina was checked through visual inspection in the morning after mating. ^3^ Females judged as having plugs were inspected later for the presence of mid-gestational fetuses. Average first litter size was determined by checking the number of fetuses isolated among the pregnant fetuses.

**Table 3 cells-09-00957-t003:** Summary of the rates of females having plugs after low-dose PMSG treatment.

Strain	PMSG Treatment ^1^	No. of Females Subjected to Mating	No. of Plugged Females ^2^	Rate (%) of Females Having Plugs
B6	No	62	16	26
B6	Yes	49	25	51
BALB/c	No	60	21	35
BALB/c	Yes	45	24	53

^1^ Females were treated with low-dose PMSG before mating with males, which is defined as “Yes”. Females were not subjected to PMSG treatment before mating with males, which is defined as “No”. In the latter case, females judged as being at the estrous stage, on the basis of visual inspection of the vagina, were selected for mating. ^2^ Presence of plugs at the vagina was checked through visual inspection in the morning after mating.

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
