# Peer review of "Modification of i-GONAD Suitable for Production of Genome-Edited C57BL/6 Inbred Mouse Strain"

_cells, 2020, doi:10.3390/cells9040957_

Round 1
Reviewer 1 Report
The manuscript is well-written. The results are also clear to show that i-GONAD is an efficient technology. Only minor comments from me. I know the authors have used PCR sequencing to check the indels. However, it would further improve the paper if the authors show the % of editing at the locus (efficiency) at the same time in Figure 2, 3 and 4.
Author Response
Reviewer 1
Answer: We are thankful for the reviewer’s cordial comments. As suggested by the reviewer, we included data on the genome editing efficiency (%) as Tables in the bottom of each Figure (see Figures 2d, 3d and 4d in the revised text)”. Concomitantly, we added phrases (shown in red) in the legend of each figure in the revised text.

Reviewer 2 Report
In this study, Kobayashi and collabs investigated the optimal current (mA) required in iGONAD delivery of RNPs for CRISPR gene editing of mice. A secondary aim of the study was to observe if application of the PMSG hormone to females prior mating increases the pregnancy success rate in C57BL6 mice. Both aims of the study are well described, designed experimentally and presented. I have no further comments for the authors.
Author Response
Reviewer 2
Thank you for revising our manuscript. The comment raised by the referee encourages our on-going study.